

# Soil nitrogen and water management by winter-killed catch crops

Norman Gentsch[1], Diana Heuermann[2], Jens Boy[1], Steffen Schierding[1], Nicolaus von Wirén[2], Dörte Schweneker[3], Ulf Feuerstein[3], Georg Guggenberger[1]

[1] Institute of Soil Science, Leibniz Universität Hannover, Hannover, 30419, Germany

[2] Leibniz Institute of Plant Genetics and Crop Plant Research Gatersleben, Corrensstraße 3, 06466 Seeland, Germany

[3] Deutsche Saatveredelung AG, Steimker Weg 7, 27330 Asendorf, Germany

*Correspondence to*: Norman Gentsch (gentsch@ifbk.uni-hannover.de)

**Abstract**

Improving N cycling in agroecosystems is one of the key challenges in reducing the environmental footprint of agriculture. Further, uncertainty in precipitation makes crop water management relevant in regions where it has not been necessary thus far. Here, we focus on the potential of winter-killed catch crops to reduce N leaching losses from N mineralization over the winter and soil water management. We compared four single catch crops (white mustard, phacelia, Egyptian clover and bristle
oat) and a fallow treatment with two catch crop mixtures with 4 and 12 plant species (Mix4 and Mix12). High-resolution soil mineral N ($N_{min}$) monitoring in combination with modelling of spatiotemporal dynamics served to assess N cycling under winter-killed catch crops, while soil water was continuously monitored in the rooting zone. Catch crops depleted the residual $N_{min}$ pools by between 40 and 72% compared to the fallow. The amount of residual N uptake was lowest for clover and not significantly different among the other catch crops. Catch crops that produce high N litter materials, such as clover and mustard
leaves, showed an early N mineralization flush immediately after their termination and the highest leaching losses from litter mineralization over the winter. Except for clover, all catch crops showed $N_{min}$ values between 18 and 92% higher on the sowing date of the following maize crop. However, only Mix12 was statistically significant. Catch crops depleted the soil water storage in the rooting zone during their growth in autumn and early winter, but preserved water later on when their residues cover the ground. The shallow incorporation of catch crop residues increased water storage capacity during the cropping season of the
main crop even under drought conditions. Hence, catch cropping is not just a simple plant cover during the winter but improved the growth conditions for the following crop at decreased N losses. Mixtures have been shown to compensate for the weaknesses of individual catch crop species in terms of nutrient capture, mineralization and transfer to the following main crop as well as for soil water management. Detailed knowledge about plant performance during growth and litter mineralization patterns is necessary to make optimal use of their full potential.



## 1 Introduction


Nitrogen (N) is one of the key elements for plant nutrition. The introduction of the Haber-Bosch process was an essential element of the "green revolution" and contributed greatly to global crop yield increases. Regardless of the sources of N from industrial fertilizers, farm manure or biofertilizers, the nitrogen use efficiency is greatest under low N input systems (Fageria and Baligar, 2005). Only 30 to50% of the applied N in high-yielding conventional agriculture is used by crop plants (Tilman

et al., 2002). A large proportion of the N is lost from agroecosystems and transferred to other ecosystems by leaching, surface runoff and volatilization. Excess N causes heavy environmental pollution, particularly in aquatic ecosystems, and creates high costs for water purification. Further, the production and application of N fertilizers requires high energy input and decreases the overall energy use efficiency of farm systems (Fuksa et al., 2013; Pimentel, 2009). Low fertilizer N recovery also increases investment costs and lowers the profitability of agricultural production. Improving N use efficiency in agroecosystems is

therefore crucial to increase the sustainability and to come closer to climate neutrality of farm systems.

Catch crops (CC, also named cover crops) are well-known tools for biological N management in arable soils of temperate climates (Thorup-Kristensen et al., 2003). Catch crops efficiently deplete residual soil mineral N ($N_{min}$) pools derived from fertilizers and crop residue mineralization and thereby lower the risk for N leaching. Between 10 and 35% of the immobilized N is thereby stored in roots and rhizodeposits (Heuermann et al. in prep., Kanders et al., 2017). The aboveground biomass

can be either harvested as fodder or act as green manure to fertilize the following cash crop and improve soil quality. In practical applications, CCs are often fertilized with farmyard manure to increase their storage capacity before winter and maximize CC biomass production. The current legal regulations in the EU allow the fertilization of CCs by up to 60 kg ha$^{-1}$ inorganic N or 120 kg ha$^{-1}$ organic N as solid manure. This generates high N loads in autumn, which have to be taken up by CCs, in addition to the N mineralized in soil.

The potential of a certain CC for soil $N_{min}$ depletion depends on the volume of their rooting zone as well as their aboveground biomass production. Dicot species, for example, were shown to deplete subsoil horizons (> 0.5 m) more efficiently than monocot species (Thorup-Kristensen, 2001). Legumes are not as effective as other plants in N depletion but have the advantage of adding substantial amounts of N from atmospheric N fixation (Thorup-Kristensen et al., 2003). Farmers' decisions for a certain CC are mainly driven to fit the crop rotation of main crops rather than to fit local weather

conditions and nutrient status of soils. Diverse catch crop mixtures have the advantage of combining several species with different rooting depths, nutrient acquisition strategies and growth optima, thus providing redundancy. Catch crop mixtures were shown to overyield the root biomass of pure stands and show higher potential for efficient soil exploration by roots (Heuermann et al., 2019). Increasing CC diversity also stimulates soil microbial diversity and biomass (Gentsch et al., 2020; Vukicevich et al., 2016), which represent a considerable reservoir for N during winter (Paul, 2007).

The advantage of winter-killed over winter-hardy catch crops is saving energy for CC termination and reducing soil tillage (Gollner et al., 2020). Even no-till practices without herbicide applications are possible if CCs are reliably terminated by frost (Romdhane et al., 2019). The challenge of winter-killed CCs, however, is the risk of losing early N by leaching and reducing

the N carry-over to the succeeding crops, if the plant residues are mineralized too fast. The C:N ratio of the CC biomass is an important measure for litter quality and its turnover during litter mineralization. In a previous study, the potential for N leaching

increased with decreasing C:N ratio, while the potential to support nutrient demands of the following crop increased (Finney et al., 2016). The study of Finney et al. (2016) also suggested that a C:N threshold exists for the aboveground biomass. At approximately C:N 20, the N retention was still at a maximum, and the crop yield service was high. Customized CC mixtures with desired litter quality might therefore be favourable for soil N management. Few studies have demonstrated the multiple ecosystem services provided by CC mixtures in terms of residual N depletion and effective green manure services (Couëdel et

al., 2018; Vogeler et al., 2019). Thus, the potential for the reduction of winter N leaching losses of CC mixtures in comparison to single species has remained poorly considered thus far.

Soil water management is becoming a new challenge with increasing aridity due to climate extremes and drought spells. Catch crops require water for their establishment and deplete the soil water pools while they grow. Competition for water with the main crop could result in yield reduction in semiarid and arid environments (Unger and Vigil, 1998). In contrast, maintaining

catch crop residues on the soil surface reduces evapotranspiration and increases the infiltration capacity of the soils. Therefore, the net effect of a CC on the soil water balance depends on the standing time, precipitation timing and capacity of the catch crop to reduce water losses (Bodner et al., 2007; Unger and Vigil, 1998).

The objectives of this study were to monitor $N_{min}$ pools and the soil water balance under different CC treatments in comparison to fallow conditions. The field trials contained monoculture vs. mixed CC stands. We followed the soil $N_{min}$ pool over 313

days from CC cultivation until the shoot elongation of the following maize crop. High-resolution soil sampling combined with modelling generated detailed insights into the winter mineralization pattern of CCs. We hypothesized that (I) the positive effects of CC on N uptake during growth as well as N mineralization after death and its transfer to the following crop could be optimized in CC mixtures and that (II) CCs do not negatively affect the soil water balance of the main crop under drought conditions.

**2 Materials and Methods**

**2.1 Field Site**

The study was conducted at a long-term field experiment in Asendorf, Northern Germany (49 m above sea level, 52°45'48.4"N 9°01'24.3"E). The soil developed from shallow loess cover over glaciofluvial sands and is classified as Stagnic Cambisol (IUSS Working Group WRB, 2014). The soil texture was silt loamy with 7% clay, 73% silt and 20% sand. The climate is

temperate oceanic with a mean average annual precipitation (MAP) of 751 mm and a mean annual temperature (MAT) of 9.3°C (long-term mean 1981-2010). The first frost usually appears at the end of November, and on the long-term annual average, 54 frost days are recorded per year. Soil monitoring spanned a 313-day period from 15[th] August 2018 to 24[th] June 2019. Temperature and precipitation during that period are displayed in Fig. S1. The year 2018 was exceptionally dry with 535 mm total precipitation, while 2019 had 779 mm.





Seven treatments in three replicates were investigated in a randomized block design with four pure catch crops, two mixtures
and a fallow with no catch crop. The pure stands were white mustard (*Sinapsis alba*), lacy phacelia (*Phacelia tanacetifolia*),
bristle oat (*Avena strigosa*), and Egyptian clover (*Trifolium alexandrinum*). A mixture of the four species was applied as Mix4
(percentage of the individual species in Table S1). For the highest species diversity (Mix12), we used a commercial 12 species
mixture (TerraLife© Maize-Pro TR Greening, Deutsche Saatveredelung, Lippstadt, Germany) composed of field pea (*Pisum*

*sativum*), crimson clover (*Trifolium incarnatum*), alsike clover (*Trifolium hybridum*), persian clover (*Trifolium resupinatum*)
and nonlegume species, including sorghum (*Sorghum sudanense*), common flax (*Linum usitatissimum*), lacy phacelia, radish
deeptill (*Raphanus sativus*), ramtil (*Guizotia abyssinica*), sunflower (*Helianthus annuus*), camelia (*Camelina sativa*), and
hungarian vetch (*Vicia pannonica*). Seeding rations and seeding densities are presented in Table S1. The catch crops followed
winter wheat (*Triticum aestivum*), and maize (*Zea mays*) followed catch crops in a two-year rotation. Catch crops were sown

on 10 August 2018 and fertilized with 47 kg N ha$^{-1}$ (UAN 28), which is 81% of the allowed N fertilization rate. Due to the
different frost limits of plants, we wanted to have equal conditions for all cover crop treatments. Therefore, CC was terminated
immediately before the first frost event (09 November 2018). This ensured equal starting condition for plant material
decomposition. Catch crops were cut just above the soil surface and left as whole plants on the ground. In spring, catch crop
residues were mulched and incorporated into the soil (maximum working depth 15 cm) two weeks before maize seeding with

a harrow/disc harrow combination. Maize was sown on 24 April 2019 and harvested on 12 September 2019 with fertilizer
applications according to Table S2. The timeline of all actions during the study is presented in Fig. S2.

**2.2 Determination of catch crop biomass and nitrogen concentration in the plant material**

Catch crop plant biomass was determined at the end of the vegetation period in three replicates per plot in 1 m wide squares.
In mixtures, plants were sorted according to the species and weight separately. Dry matter yield was determined gravimetrically

after drying of plant material at 60°C until constant weight. For root biomass quantification, three soil cores were taken per
plot, having a diameter of 6 cm and a length of 100 cm. Soil cores were cut into three pieces, representing 0-30 cm, 30-60 cm
and 60-100 cm soil depth. Buckets with a middle-placed high cylindric pipe, opened at the top and equipped with a 0.4 mm
mesh-sized web at the bottom, and a continuous water flow around that pipe were used to elutriate the samples. During this
procedure, lightweight roots separated from heavier soil particles and could be captured in the web at the bottom of the cylindric

pipe while water run over the top end of the pipe. Additional lightweight particles like seeds or straw were sorted manually
from ten randomly taken samples per soil depth. In those samples, roots were completely separated from remaining material.
After drying root material at 60 °C to constant weight, dry mass could be determined. A mean value of 67.3 % of non-root
material dry weight (mainly straw from pre-grown wheat) was applied to all samples in 0-30 cm depth, while in the deeper
soil layers shares of non-root material were neglectable. Total organic carbon (OC) and N (TN) concentrations in plant

materials were measured with dry combustion in an elemental analyser (EuroEA3000, Hekatech, Wegberg, Germany).



### 2.3 Soil mineral nitrogen and volumetric water content

Soil samples were taken with a manual soil corer from four soil depths (0–10, 20–30, 50–60, and 70–80 cm). Soil samples were immediately stored in a cooler and transported to the laboratory for fresh sample extraction. Approximately 10 g of soil material was extracted with 40 ml of a 0.0125 M $CaCl_2$ solution after shaken overhead for 60 min. Soil $N_{min}$ was determined as the sum of the measured $NH_4^+$ and $NO_3^-$ concentrations by an Autoanalyzer (SAN-plus, Skalar Analytical B.V., Breda, The Netherlands). A sample aliquot was dried at 105°C to determine the gravimetric soil water content (% dry mass) and relate $N_{min}$ values to the dry weight base. Bulk density (BD) was determined gravimetrically after drying undisturbed soil cores (100 cm$^{-3}$) at 105°C. Soil cores were taken by a stainless core cutter (100 cm$^{-3}$) only once per plot, and we assume that the data did not change significantly during the observation period. Field capacity was estimated from pedotransfer functions using soil texture and BD according to Ad-hoc-AG Boden (2005). The volumetric soil water content (vol%) was calculated on the soil mass base from the gravimetric values. Total water storage was calculated as the sum of all soil depth increments and referred to the total soil volume to 80 cm depth. $N_{min}$ stocks were calculated for individual soil depth increments (interpolation see section 2.4) with Eq. (1) and summarized to 80 cm soil depth:

$$N_{min}\ (kg\ ha^{-1}) = \frac{N_{min}\ (mg\ kg^{-1}) \times BD\ (g\ cm^{-3}) \times depth(cm)}{10} \tag{1}$$

### 2.3 Data logger

As a complement to our offline soil moisture measurements, we installed data loggers for continuous monitoring. We used CS650 multiparameter sensors connected to a CR300 data logger (Campbell Scientific, Inc., Logan, USA). The logger delivered hourly data on soil volumetric water content, temperature and bulk electrical conductivity for the upper 0 cm to 30 cm soil depth. In the following, we discuss the data on volumetric water content only. The loggers collected continuous data for one year (from 27 September 2018 to 24 September 2019) with short removals for soil preparation. Unfortunately, we had serious damage from wild animals and lost six replicates for statistical evaluations.

### 2.4 Statistics and depth-time visualization

In total, 840 $N_{min}$ samples at 10 time points were incorporated in the spatiotemporal visualization of the values (time points are shown in Fig. S2 and the primary data files). In the first steps to produce heat maps (Fig. 2 to 4) with R version 4.0.2, field replicates (N=3) were summarized per treatment for easier data handling (R Core Team, 2020). Next, horizontal interpolation was performed on a daily basis by a local polynomial regression model (span = 0.2). The same model was applied for depth interpolation on 50 equally distributed data points until 80 cm soil depth. With this procedure, we generated 15700 data points per treatment and plotted them with a geom_raster in ggplot2 version 3.3.2 (Wickham, 2016). Differences in plant biomass and OC and TN concentrations were analysed by pairwise t-tests (pairwise test package). Values for total plant OC and TN concentrations and C:N ratios were calculated as weighted means according to the equivalent mass ratio of roots and shoots. We used bootstrapping prior to pairwise t-tests to compare $N_{min}$ stocks between treatments at specific sampling dates. All



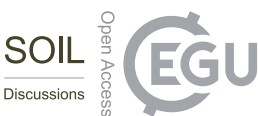

metadata and R-scripts can be publicly accessed from the DOI prowided below (Sect. Code and data availability). All statistical
statements, plots and calculations of data variability can be reproduced from the scripts.

## 3 Results

### 3.1 Biomass and plant nitrogen accumulation

Total shoot biomass was highest for phacelia, followed by oat, the two mixtures and mustard (Table 1). Clover showed less
than half of the shoot biomass compared to the other CC treatments. Root biomass summarized to 90 cm soil depth was highest
for oat and decreased in the order Mix12, Mix4, phacelia, mustard and clover. Total plant biomass followed a similar pattern.
The highest average biomass N was found in Mix12, but differences to other treatments were not significant due to large data
variability (Table 1). Despite the low biomass production, clover produced remarkably high biomass N, in the same range as
mustard. The biomass C:N ratios clearly distinguished the different CC treatments and total C:N ratio increased in the order
clover, Mix12, oat, mustard, phacelia and Mix4. C:N ratios were significantly higher in roots than in shoots, and the lowest
root C:N ratio was found in Mix12. In total, the proportion of legumes in the shoot biomass of mixtures ranged from 0.5 to 1%
in Mix4 ($1.03 \pm 0.16\%$ on average) and 7 to 22% in Mix12 ($16.2 \pm 2.6\%$ on average), and thus explained the lower C:N ratio
of Mix12 as compared to Mix4.

### 3.2 Soil water budget

Calculations of the total soil water content showed that in all depth increments, the field capacity of the soil was never exceeded
in the soil profile (Fig. S3). A period of higher rainfall starting at the end of January (Fig. S1) that increased the soil water
content in the upper 10 cm to 66 to 95% of field capacity.

Total soil water storage was calculated from the summarized volumetric water content to 80 cm soil depth and is referred to
the mean values of fallow as a reference (100%) in Fig. 1. The periods of CC growth clearly deplete the soil water storage
until its maximum growth. At the beginning of November 2019, relative moisture levels under CC reached their minimum,
between 17 and 33% below the water storage of fallow, which was significant for all CCs except clover (Table S3). Already
before CC termination, the soil water storage started to recover and increased towards the fallow level until mid-December
2018 (Fig.1). Thereafter, all CC treatments showed significantly higher soil water storage than the fallow treatments (Table
S3). After soil preparation and maize sowing, all CCs still showed higher soil water storage than the fallow CCs (mustard
+4%, clover + 5%, oat +6%, phacelia +12%, Mix4 +14%, Mix12 +9%), but only phacelia and Mix12 were statistically
significant ($p < 0.05$).

The loggers delivered one year of continuous data of the volumetric water content to 30 cm soil depth and supported the results
from our soil sample measurements. Due to the loss of replications, we were not able to apply proper statistical comparisons



**Table 1: Plant biomass data of shoots, roots and total biomass from different catch crop treatments. Mean values of six plots per treatment and standard error (SE) are shown. Small letters (Sig) represent contributions to significantly different groups at p < 0.05.**

| Catch crop | Biomass (t ha$^{-1}$) | | | TN (%) | | | C:N ratio | | | Biomass N (kg ha$^{-1}$) | | | Root:shoot ratio | | |
|---|---|---|---|---|---|---|---|---|---|---|---|---|---|---|---|
| | Mean | SE | Sig | Mean | SE | Sig | Mean | SE | Sig | Mean | SE | Sig | Mean | SE | Sig |
| | | | | **Shoot** | | | | | | | | | | | |
| Mustard | 4.16 | 0.23 | a | 2.28 | 0.16 | ab | 19.31 | 1.36 | ab | 96.35 | 11.61 | ab | | | |
| Clover | 1.94 | 0.17 | c | 4.36 | 0.05 | c | 10.24 | 0.12 | c | 83.91 | 7.38 | b | | | |
| Oat | 4.95 | 0.37 | ab | 2.57 | 0.39 | ab | 18.70 | 1.83 | ab | 125.99 | 18.34 | ab | | | |
| Phacelia | 5.34 | 0.39 | b | 2.04 | 0.10 | b | 20.13 | 1.14 | b | 107.65 | 5.86 | a | | | |
| Mix4 | 4.95 | 0.14 | b | 1.98 | 0.07 | b | 21.60 | 0.74 | b | 98.35 | 5.36 | ab | | | |
| Mix12 | 4.29 | 0.20 | a | 2.60 | 0.12 | a | 15.96 | 0.68 | a | 111.15 | 5.16 | a | | | |
| | | | | **Root** | | | | | | | | | | | |
| Mustard | 5.03 | 0.82 | a | 1.28 | 0.10 | ab | 30.00 | 1.28 | bd | 66.01 | 12.59 | a | | | |
| Clover | 4.76 | 0.83 | a | 1.51 | 0.05 | b | 27.51 | 0.89 | abc | 73.57 | 13.94 | a | | | |
| Oat | 12.42 | 1.11 | b | 1.01 | 0.04 | c | 32.36 | 1.16 | d | 124.47 | 10.92 | b | | | |
| Phacelia | 7.18 | 1.43 | a | 1.34 | 0.07 | ab | 26.70 | 0.41 | ac | 96.71 | 19.59 | ab | | | |
| Mix4 | 7.59 | 2.20 | ab | 1.31 | 0.07 | a | 28.41 | 0.81 | bc | 103.52 | 35.76 | ab | | | |
| Mix12 | 11.15 | 2.75 | ab | 1.54 | 0.09 | ab | 25.47 | 0.51 | a | 179.33 | 48.82 | ab | | | |
| | | | | **Total** | | | | | | | | | | | |
| Mustard | 9.20 | 0.91 | ac | 1.76 | 0.12 | ab | 23.57 | 1.40 | ab | 162.35 | 19.31 | a | 2.59 | 0.53 | bc |
| Clover | 6.70 | 0.83 | c | 2.32 | 0.09 | c | 18.93 | 0.51 | c | 167.80 | 12.74 | a | 1.21 | 0.19 | a |
| Oat | 17.37 | 0.83 | b | 1.45 | 0.08 | b | 25.09 | 1.19 | b | 250.47 | 18.65 | b | 2.68 | 0.46 | b |
| Phacelia | 12.52 | 1.48 | a | 1.66 | 0.06 | b | 22.80 | 0.83 | ab | 204.35 | 17.81 | ab | 1.38 | 0.30 | ac |
| Mix4 | 12.54 | 2.13 | ab | 1.63 | 0.08 | b | 24.72 | 0.97 | b | 201.87 | 34.70 | ab | 1.58 | 0.48 | abc |
| Mix12 | 15.44 | 2.92 | ab | 1.89 | 0.04 | a | 21.10 | 0.59 | a | 290.48 | 50.66 | ab | 2.50 | 0.51 | abc |



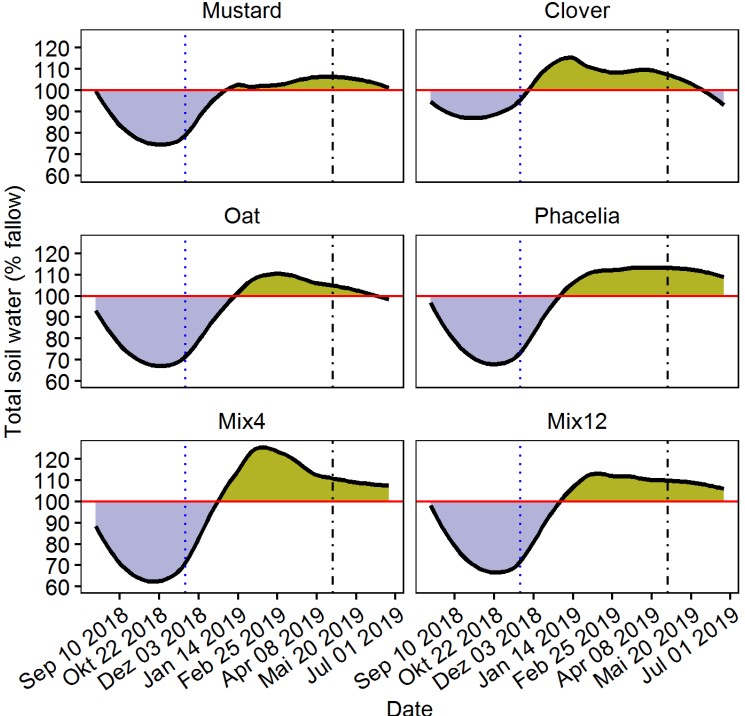

**Figure 1: Volumetric soil water content down to 80 cm soil depth referred to fallow as 100% reference (red horizontal line). Blue areas mark soil water content below the fallow level and yellow/brown values above. Continuous black lines were achieved by logistic regression models from three replicates, and differences between the treatments at the individual sampling dates are provided in Table S3. The blue dotted line marks the termination of CC, and the vertical dashed line marks the seeding date of the main crop maize.**

of treatments. Nevertheless, the data fit the measurements of the volumetric water contents in the laboratory and allowed us to extend the interpretation beyond maize harvest. In agreement with the laboratory measurements, the logger data showed depletion of soil water by CC of all treatments (Figs. S4 and S5). Soil water was restored until soil preparation for maize seeding, and exceeded the fallow levels except for mustard (Fig. S4; mustard ± 0%, clover + 13%, oat + 9%, phacelia +12%, Mix4 +13%, Mix12 + 14%). During maize growth, all CC treatments, except for Mix4, where we lost all replicates, exceeded the fallow level. Particularly with the start of August 2019, when maize reached the stage of corn filling, the CC treatments exceeded the fallow by up to 200% (Fig. S5).

**3.3 Soil N depletion and temporal N$_{min}$ fluctuations**

In Figs. 2 to 4, the dynamics of N$_{min}$ concentrations can be followed in temporal heat maps as model results from individual sampling points. After the exceptionally dry summer of 2018, the onset of precipitation in combination with mild weather generated high autumn N mineralization. This mineralization pulse became particularly relevant in the fallow treatments up to



a soil depth of approximately 40 cm between the start of September and mid-November (Fig. 2.a). Below this depth, the soil was still $N_{min}$ depleted from the previous wheat crop uptake. After the first frost periods (Fig. S1), the mineralization pulse decreased, and precipitation events resulted in descending soil water migration accompanied by diffuse $N_{min}$ leaching toward

the subsoil. In the first half of December, the subsoil $N_{min}$ concentrations at 70 cm to 80 cm under fallow increased significantly (p = 0.0034) by threefold (from $1.1 \pm 0.2$ to $3.4 \pm 0.3$ mg $g^{-1}$) and reached their maxima in January 2019 ($8.7 \pm 0.7$ mg $g^{-1}$). Thereafter, the $N_{min}$ concentrations at 70 cm to 80 cm decreased again until the end of the observation ($4.5 \pm 0.5$ mg $g^{-1}$). The fluctuations in N mineralization gains and N leaching losses were very prominent at 20 cm to 30 cm depth under fallow conditions. Here, the $N_{min}$ values dropped from a maximum of $19.3 \pm 1.6$ mg $g^{-1}$ in November until $1.9 \pm 0.6$ mg $g^{-1}$ at the last

sampling before maize seeding and fertilization.

Clover was the only CC treatment that showed a similar pattern as bare fallow (Fig. 3b). The variability between the maxima and minima, however, was not as strong as in the fallow. Despite this, clover reduced the average $N_{min}$ stocks in November 2018 by 40% compared to the fallow stocks, but the variance of the data was too large to be significantly different (Fig. 5, Table 2). All other CC treatments significantly reduced the $N_{min}$ pools in autumn by 66% to 72% compared to the fallow

treatment (Fig. 5, Table 2). Mustard reduced the autumn mineralization pulse by $80.3 \pm 6.9$ kg $ha^{-1,}$ but unlike phacelia and oat, for example, we observed a very early increase of higher $N_{min}$ loads from litter mineralization. Under mustard, the $N_{min}$ concentrations in the upper 20 cm increased strongly from $4.0 \pm 0.4$ mg $g^{-1}$ at termination date to $13.8 \pm 2.4$ mg $g^{-1}$ at early December. This represents an increase in the $N_{min}$ stocks by 17.9 kg $ha^{-1}$.

The quantification of the N leaching from downward migration with the hydraulic gradient was not straightforward. However,

we could estimate the leaching from fallow and clover by simple calculation of the difference from $N_{min}$ maxima and minima during the winter. We calculated a total N leaching to > 80 cm soil depth of 103 kg $ha^{-1}$ for the fallow and 81 kg $ha^{-1}$ for the clover as the sum over three periods (calculation provided in the R files). Considering 47 kg N added by fertilization to the fallow, we end up with 54 kg $ha^{-1}$ for fallow and 34 kg $ha^{-1}$ for clover derived from mineralization of organic N pools throughout the winter. For the other CC treatments, we recorded a constant increase in total $N_{min}$ stocks in the soil profile (0-80 cm) over

time. Therefore, it was not possible to simply summarize gains and losses. Instead, the gains and losses in the 70 cm to 80 cm soil depth were used as indicators of $N_{min}$ discharge. The calculated $N_{min}$ losses were 7.4 kg $ha^{-1}$ for phacelia, 6.0 kg $ha^{-1}$ for oat, 18.2 kg $ha^{-1}$ for Mix4, and 10.0 kg $ha^{-1}$ for Mix12. For mustard, we assume a constant in and output at 70 cm to 80 cm soil depth. The discharge rate was calculated from a linear regression to 0.11 kg $ha^{-1}$ $day^{-1}$, and the summarized $N_{min}$ loss over 156 days was 18.4 kg $ha^{-1}$.

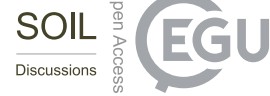

**Figure 2: Spatiotemporal resolution of N$_{min}$ concentrations in soil spanning a 313-days period under a) fallow, b) mustard, and c) clover. The dashed line marks the seeding of maize following catch crops or fallow.**




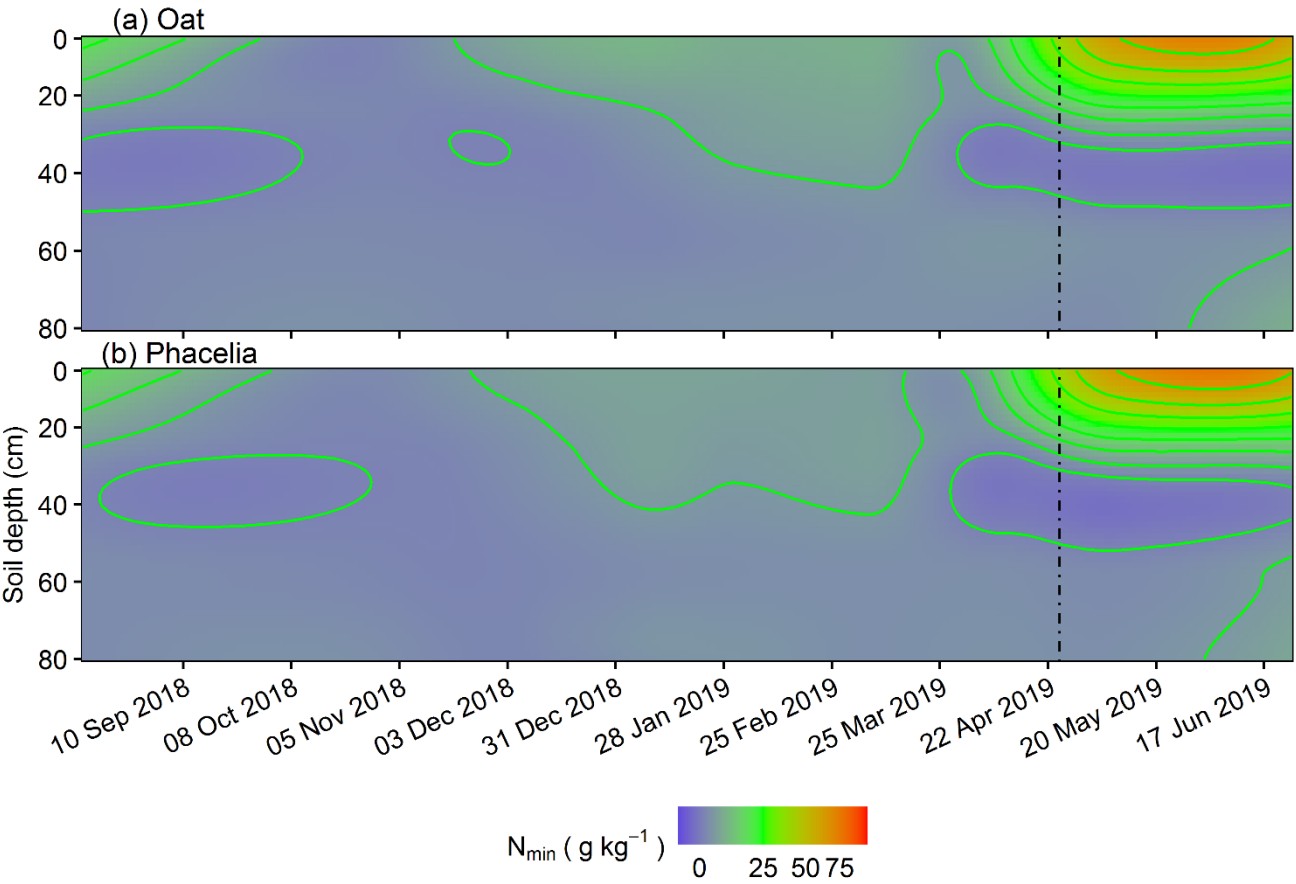

Figure 3: Spatiotemporal resolution of $N_{min}$ concentrations in soil spanning a 313-days period under a) oat, b) phacelia. The dashed
line marks the seeding of maize following catch crops.



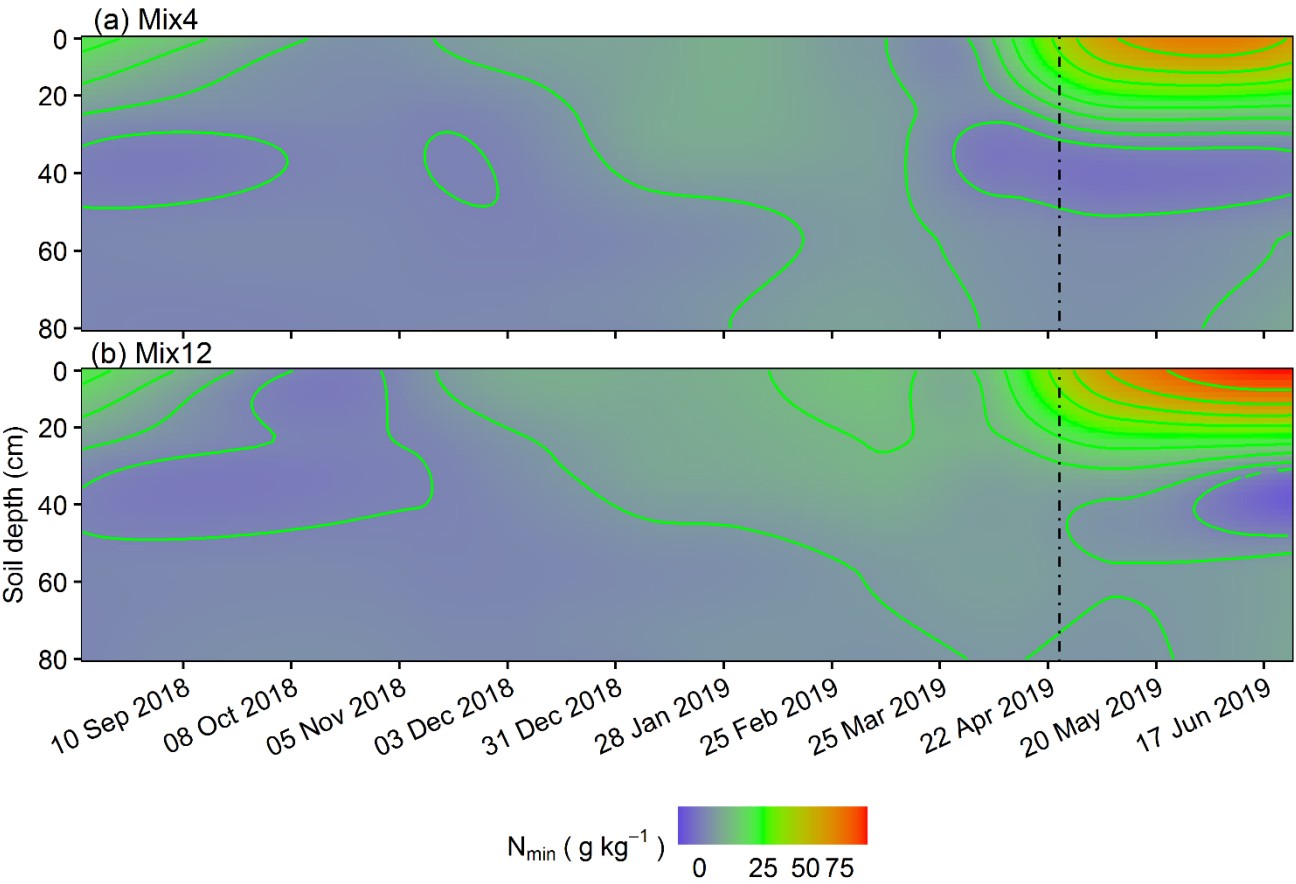

**Figure 4: Spatiotemporal resolution of $N_{min}$ concentrations in soil spanning a 313-days period under a) Mix4 and b) Mix12. The dashed line marks the seeding of maize following catch crops.**





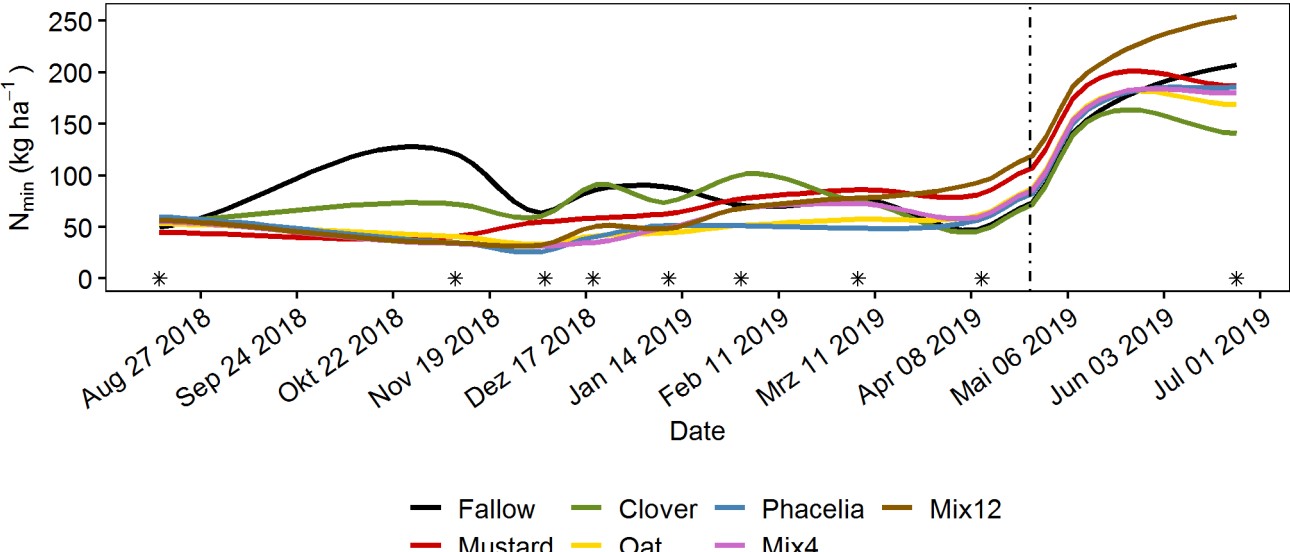

**Figure 5: Development of soil mineral N stocks ($N_{min}$ in kg ha$^{-1}$) during the observation period under different CC treatments. Each line represents the mean of three replicates. The lines were achieved by logistic regression models from nine sampling dates and tree replicates per treatment. Differences between the treatments at the individual sampling dates are provided in Table 2 (marked by asterisks). The dashed line marks seeding of the main crop maize and onset of N fertilizer application.**

**Table 2: Pairwise comparison of soil mineral N stocks ($N_{min}$ in kg ha$^{-1}$) from Fig. 5 at the individual sampling dates. R codes and data are provided in the supplement. Small letters denote the contribution to significantly different groups.**

| Catch crop | 15 Aug 2018 | 09 Nov 2018 | 05 Dec 2018 | 19 Dec 2018 | 10 Jan 2019 | 31 Jan 2019 | 06 Mar 2019 | 11 Apr 2019 | 24 Jun 2019 |
|---|---|---|---|---|---|---|---|---|---|
| Fallow | a | a | a | bc | b | a | ac | c | a |
| Mustard | a | b | a | ab | ab | a | a | ab | a |
| Clover | a | ab | Ab | c | ab | a | abc | c | a |
| Oat | a | b | Bc | ad | a | bc | bc | c | a |
| Phacelia | a | b | C | ad | a | b | b | c | a |
| Mix4 | a | b | C | d | a | ac | abc | ac | a |
| Mix12 | a | b | Bc | ad | a | ac | a | b | a |

## 4 Discussion

### 4.1 Soil water budget

Catch crops deplete the available soil water during their growth (Fig. 1, Fig. S5). At the same time, the convective transport rate of pore water throughout the soil column is lower as under bare fallow. The lower seepage water rate causes a reduction of N leaching losses in humid years. The extreme drought throughout 2018 did not result in seepage during the winter of



2018/2019, as winter precipitation was not enough to restore the field capacity of the soils (Fig. S3). Thus, the convective transport of water by the mass flow through the soil column was absent. Despite the missing seepage, our data indicate $N_{min}$

movements throughout the soil profile, caused by the hydraulic gradient and diffusive N transport by concentration gradients as principle drivers (Cameron et al., 2013). We can exclude capillary rise from the subsoil, as the groundwater table was > 3 m below the surface. The much drier subsoils than surface soils in winter 2018/2019 (Fig. S.3), causing a strong hydraulic gradient toward the subsoil.

Our data showed that in mid-December 2018 (4 weeks after CC termination), the soil water budget of CC was equal to the

fallow budget and exceeded their level thereafter (Fig. 1). We assume the combination of several factors as drivers for constantly higher soil water content by all catch crop variants. (I) The exposed bare soil of the fallow is prone to splash effects by precipitation. Soil pores can be clogged, and surface crust formation restricts water infiltration to the soil, causing surface runoff. Catch crop residues increase soil roughness, protect the soil surface from splash effects and minimize surface runoff (Unger and Vigil, 1998). (II) Catch crops produce biopores by decaying roots. Preferential flow paths following these root

channels (Kautz, 2015) and increases water infiltration. (III) The litter of frost-killed CCs acts as a mulch and reduces surface dry out by wind and interception losses (Unger and Vigil, 1998; Bodner et al., 2007, 2015).

However, not all CC species are equally suitable for water conservation. Species vary in their transpiration rates, water use efficiency, rooting depth or canopy coverage. Bodner et al. (2007) demonstrated that phacelia and vetch had a substantially lower evapotranspiration than rye and mustard. Within their study, mustard showed by far the largest transpiration rates. Our

data indicate that clover consumed only half of the water during its growth compared to the other CCs. This was due to the lower biomass and rooting depth (Table 1; Heuermann et al., 2019). The soil of the upper 0 to 30 cm following clover constantly showed a 10 to 33% higher water content than fallow during the whole maize cropping period (median 19%, Fig. S5). Mix12 and Phacelia showed similar results until mid of July. When maize plants entered the generative stage, all CC treatments accept of clover showed up to 200% higher soil water contents than the fallow treatments (Fig. S5). Similar results of higher crop

water availability for the following crop were reported by Kaye and Quemada, (2017) and explained by a stronger mulch effect and higher soil organic matter (OM) contents. Interestingly, in our experiments only clover did not show values about 33% during the generative maize stage. The only explanation we came up with thus far could be the lower root and shoot biomass by clover that produced less particulate OM and biopores compared to the other CC treatments. This could result in a lower mulch effect and slower infiltration in the upper 30 cm of the clover treatments. The reasons that caused the differences between

clover and other CC treatments remain quite speculative. Infiltration experiments, measurements of evaporation rates, aggregate fractionation or evaluation of biopores could help future studies explore the potential of CCs for soil water management.

Overall, catch crops deplete soil water while they are growing and preserve water when they are killed, and their residues cover the ground. Catch crops can thus be a part of the tools for soil water management. In drought years, early frost killing or

termination by rolling, crimping or mulching will stop transpirative water losses from the soil. Further, the residual mulch will decrease evaporation and increase infiltration. In moist or normal years, winter hardy catch crops and late termination can be



used to dry the soil to a level that is optimal for seed bed preparation (Kaye and Quemada, 2017). Phacelia and Mix12 showed the most consistent results concerning the temporal course of the water contents. We thus conclude that catch crop mixtures with high diversity combine several positive effects for soil water management for the following crop.


## 4.2 Soil N management with winter-killed catch crops

This study clearly visualizes $N_{min}$ migration in soil following drought. Although there was only slow, unsaturated soil water migration, $N_{min}$ leaching toward the subsoil was recorded as a result of the hydraulic gradient. Our calculations indicated that the fallow lost at least 54 kg ha$^{-1}$ $N_{min}$ derived from residual fertilizers of the preceding crop and the autumn mineralization

pulse of crop residues and soil organic matter. In wetter years with high seepage rates, $N_{min}$ leaching of up to 80 kg ha$^{-1}$ from the fallow at same sites as been recorded (data not shown).

Clover reduced the soil $N_{min}$ stocks in autumn by 40%. This was only half of the reduction as by the other CC treatments and well in the range from similar studies (Couëdel et al., 2018). The low biomass and thus N uptake and the clover's lower dependence on N fertilizers due to N fixation are the main reasons for the weak performance on residual N depletion. Therefore,

the current legislation does not recommend the fertilization of legumes. Winter hard varieties such as winter vetch (*Vicia villosa*) or white clover (*Trifolium repens*) and/or inclusion in mixtures of various legume species can extend the N mineralization pulse to the main crop season. Legumes perform several ecosystem services other than residual N depletion. Nitrogen facilitation, the transfer of N from legumes to companion plants in mixtures (Paynel and Cliquet, 2003), is crucial to maximizing CC biomass without fertilizer application and reducing interspecific competition (Duchene et al., 2017). Legumes

in cropping systems reduce the total amount of energy that is required for crop production by 12% to 34% (Jensen et al., 2012). Furthermore, legumes are integral parts of complex soil microbial abundance and diversity, the regulation of litter quality by C:N ratios (Table 1), the improvement of microbial C use efficiency and the acceleration of soil OM build-up (Jensen et al., 2012).

For mustard, we observed a very early flush of $N_{min}$ from litter mineralization immediately after termination (Fig. 2b). In

January 2019, we measured similar $N_{min}$ concentrations in 80 cm soil depth compared to fallow or clover. Although the C:N ratios of roots (30.0) and total shoots (19.3) did not differ significantly from phacelia and oat (Table 1), previous studies showed that the C:N ratio of mustard leaves alone is between 8 and 11 (mean 10.5; Table S4). The high N leaf litter material is most likely the main cause for the early mineralization flush and contributes to the higher $N_{min}$ leaching potential from mustard residues over the winter. The fibre-rich mustard stalks, however, showed C:N ratios between 23 and 41 (mean 34)

and were leftovers on the soil surface after the winter (Fig. S5). The incorporation of low N catch crop materials is likely to cause microbial N immobilization with negative impacts on the main crop N supply and crop yield services at a C:N ratio of 25 (Finney et al., 2016). However, during the initial maize growth, we did not detect a pattern of N immobilization for mustard and measured even the highest N loads together with Mix12.





The lowest $N_{min}$ losses were calculated for the pure stands of phacelia and oat. Oat roots had wider C:N ratios (Table 1 and

S3), and the shoot litter remained as a thick litter mat on the soil surface (Fig. S6). This most likely extended oat litter mineralization to the early spring. Phacelia showed a similar N mineralization pattern as oat (Fig 3), but the C:N ratios were quite narrow, particularly in phacelia roots (Table 1 and S4). It is still unclear why the decomposition of phacelia litter is similarly retarded as for oat, as litter quality is unlikely to explain this observation.

The mixing of different catch crop species can compensate for the weaknesses of individual single catch crops with respect to

residual N depletion, extension of winter mineralization, and N transfer towards the following maize crop. We demonstrated that CC mixtures with up to 22% legumes in shoot biomass took up similar amounts of $N_{min}$ during the growing period from soil compared to pure non-legume CCs. Similar effects of mixed CCs were previously reported by Couëdel et al. (2018). At the same time, the N fixation service by legumes improved litter quality in Mix12 with low C:N ratios in shoot and root litter. Lower C:N ratios of litter materials have been found to respond to higher carbon-use efficiency from the microbial community

(Manzoni et al., 2012), which triggers OM build-up in soil (Nicolardot et al., 2001; Liu et al., 2018). Mix4 did not perform well in terms of litter quality adjustment. Egyptian clover, which is adopted to Mediterranean climate, was not competitive enough to contribute approximately 1% to the total shoot biomass, so that mustard and phacelia dominated the mix. Care must be taken to ensure a suitable balance between companion plants in low species mixtures to achieve the desired functions. Nevertheless, Mix4 did not show a very early winter mineralization pulse, such as mustard and clover allone, and extended

the recovery of $N_{min}$ pools until January 2019.

The $N_{min}$ supply to the following crop was best for Mix12 and 92% higher than the fallow level at the time of maize seeding, which was also confirmed from previous studies on CC mixtures. Farneselli et al. (2018), for example, found that vetch best supported the nutrient demand of processing tomatoes, but resulted also in high N leaching from the soil. In contrast, vetch mixed with barley ensured adequate tomato nutrition but minimized leaching losses. Rinnofner et al. (2008) found that

mixtures of legumes and non-legumes performed best in terms of a balanced biomass N yield, biological N fixation, and soil N depletion compared to pure legumes or non-legumes. Crucifer-legume mixtures mineralized more N and had a higher green manure effect on the following crop than pure crucifer CCs (Couëdel et al., 2018). However, CC mixtures did not necessarily perform better than their best-performing constituent monoculture. Florence and McGuire (2020) analysed seven metrics on CC performance in a review of 27 studies. They found that the best monocultures and mixtures perform comparably if the

metrics are compared independently to each other. The true potential of CC mixtures is, however, their multifunctionality. In our study, each of the four monocultures showed specific weaknesses in terms of biomass production, residual N-uptake, N-leaching, N-mineralization and delivery toward the following maize crop and soil water conservation. In all of this metrics, mixtures and particularly the high diversity Mix12, always performed comparable to the best-performing monoculture and were the best compromise in terms of multifunctionality. Therefore, we conclude that mixtures can compensate individual

weaknesses of monocultures and provide functional redundancy. Overall, high-diversity CC mixtures have strong agronomic potential for optimizing N cycling and crop water management in arable soils.





## 5 Conclusion

The depletion of the soil water pool can have negative impacts on the water availability of the following crop and their yields if the deficit cannot be restored. Thus, some authors suggest that catch crops and green manures are not suitable for dry regions

(Unger and Vigil, 1998). However, crop production must face the challenge of climate change. In central Europe, the extreme drought during 2018 and 2019 triggered a dramatic water deficit with disastrous negative impacts on crop yields. Such extreme events are projected to increase in the near future, and farming systems have to deal with a higher uncertainty in crop water supply (Spinoni et al., 2018). Active soil water management has become a new task for crop production in regions where it has not been relevant thus far (Bodner et al., 2015). Under such extremes, the major concern of farmers is that water use by

catch crops will reduce the water for the following crop.

This study demonstrates that catch cropping during dry years did not result in water shortages for the following crop. Catch crops deplete soil water while they are growing but reduce evaporation and preserve water compared to bare fallow after their dieback. The shallow incorporation of CC residues into the soil increased the infiltration and water storage capacity under the following main crop. The CC biomass and the percentage of soil cover from their residues determine the magnitude of the

benefits for the following crop. Phacelia and Mix12 showed the most consistent combined results from data loggers and soil moisture measurements in the laboratory. High diversity mixtures most likely combine several positive aspects of soil water management that result from an optimized rooting depth and volume, plant transpiration, soil coverage by mulch and evaporation, and OM input.

All CC treatments depleted the soil $N_{min}$ pools while they grew. The magnitude of depletion depends on the standing biomass

and the rooting volume. Our results demonstrated that even mixtures up to a percentage of 22% legumes in the biomass are not limited in their N uptake efficiency compared to non-legume CCs. Winter mineralization of frost-killed CCs can produce variable loads of $N_{min}$ in the soil profile. The starting point of N mineralization is controlled by the first frost that killed the CC or by the induced termination. In particular, clover and mustard produced high N litter materials (specifically mustard leaves) that are undergoing fast degradation. These two CCs were those with the highest $N_{min}$ losses until the next vegetation period.

Additionally, Mix4 with 50% mustard in litter showed similar $N_{min}$ leaching losses as mustard, but the time point was extended by approximately 6 weeks toward the new seeding. Litter quality in terms of low C:N ratios was not always a good explanation for fast N mineralization. Phacelia had low C:N ratios, particularly in roots and leaves, but indicated a low winter N mineralization potential and leaching losses. Future studies could explore the potential of phacelia for substances that inhibit the N mineralization cycle. With respect to the investigated parameters of the N cycle, CC mixtures always performed

comparable to the best-performing monoculture and compensate individual weaknesses of monocultures. In order to minimize leaching losses from CC mineralization, future studies on CC mixtures should therefore consider their litter N mineralization potential. It might also be advisable to apply a certain percentage of frost hardy CC that could be easily terminated right before seedbed preparation. The adjustment of CC mixtures to the following crop is a challenge for seed suppliers. The potential of single catch crops to support crop rotation diseases must be taken into account, as well as the nutrient demand of the following

crop. The applied Mix12 was specifically designed as a pre-crop to maize. In this example, the mineralization of Mix12 litter fit the continuously rising N demand of maize plants well, particularly in the later stages, when the fields cannot be accessed anymore. Manipulating the plant diversity of CC rotations can be a promising approach to optimize N cycling and minimize leaching losses from agroecosystems.

**Code and data availability**

All metadata and R codes to reproduce the content of this study are publicly available under the Creative Commons Attribution 3.0 Germany. The files are accessible from the Zonodo achieve by the following DOI: https://doi.org/10.5281/zenodo.5603221.

**Author contributions**

NG was planning the research activity, participated in fieldwork and provided the statistic evaluation and R codes. SS, DH 400 and DS was participating in field and laboratory work and provided metadata. JB, NW, UF and GG was involved in conceptualization, funding acquisition, project coordination and supervision. NG was preparing the manuscript with contributions from all authors.

**Competing interests**

The authors declare that they have no conflict of interest.

**Acknowledgements**

This work is part of the BonaRes (Soil as a Sustainable Resource for the Bioeconomy) project CATCHY (Catch-cropping as an agrarian tool for continuing soil health and yield increase) funded by the German Federal Ministry of Education and Research (BMBF), project number 031A559C. We grateful to Silke Bokeloh and the whole laboratory team from the institute of soil science for assistance with sample preparation and measurements.

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
