# Peer review of "Soil nitrogen and water management by winter-killed catch crops"

_SOIL, 2021_

## Author Response (AR2)

**Respond to Reviewer #1**

We thank the reviewer for the helpful comments that helped to improve the quality of the manuscript further. In the following the reviewer's comments are marked in italic followed by our response.

*The aim of the manuscript is clear and the hypotheses are well defined.*

*The Introduction provides an actual and sufficient literature review. I only recommend mentioning the importance of catch crops in protecting the soil from water and wind erosion.*

Done.

*L52: "Legumes are not as effective as other plants in N depletion…" - You could add reasons.*

We add the reasons.

*The chapter Materials and Methods is clear and detailed. However, I miss parcel size information (L95) and the way of sowing (by hand/by seeder/on surface/depth of sowing; L105). You should also provide information about the method of soil tillage after wheat harvest. Provide also the date of the catch crop plant biomass determination (L113).*

The information were all added to the materials and methods section.

*L127: "Soil samples were taken…" – When? How many times? Does the Figure S2 provide the data?*

The following sentence was added:

The sampling campaign comprised 9 sampling points in the period from 15$^{th}$ of August 2018 to 24$^{th}$ of July 2019 (Fig. S2, all dates provided in Table 2).

*L148: "...samples at 10 time points…" – 10 time points or 9 time points (according to black dots in Figure S2)?*

9 is correct.

*L93: Consider adding a table with the monthly precipitation values, the Figure S1 is less clear.*

I added a new Table S5.

*The Results are presented clearly and adequately described. The Discussion is also processed at a very good level. I only recommend adding information about volunteer winter wheat and weeds.*

I added the following paragraph to the results:

The slow biomass development of clover encouraged winter wheat and several regional weeds to germinate. These were weeded manually once. The rest of the CC treatments

suppressed weeds very well. Growth of weeds and volunteer winter wheat was documented and their contribution to total biomass was negligible.

*Technical corrections:*

*L132 and 133: 100 cm-3 = 100 cm3*

Changed

*Figure 2 to Figure 4: You provide units g kg-1 for the figures but mg g-1 in the text (L211 to L222)*

Changed to g kg-1.

*L216: "Clover…as bare fallow (Fig. 3b)." = Fig. 2*

Changed to Fig.2

*Table 2: In the column "05 Dec 2018" are not only small letters*

That was a mistake probably from auto-correction issues.

*Figure 5, Figure S1 and Figure S2: You should use the English names (abbreviations) of the months.*

Sorry, changed.

*References: Check notation of literature sources (e.g. missing journal titles, abbreviated/full journal titles)*

There might be an issue with the citation style in my Zotero. But this will be clarified with the production office in terms of acceptance. Probably they can provide me a correct citation style file. Otherwise I need to correct manually.

**Respond to reviewer #2**

We thank the reviewer for the constructive comments, which helped to improve the manuscript quality further. In the following the reviewer comments are printed in italic followed by our response in plain text.

*Specific comments:*

*All CC were fertilized, including clover and the mixtures (containing legumes). Based on recommendations and previous studies, N fertilization was not only not needed, but can be expected to have affected the expression of species in the mixtures (see, for example, https://doi.org/10.1371/journal.pone.0235868 and https://doi.org/10.1016/j.agee.2020.107287). In addition, you terminated all CC at the same time, which makes the comparison between different species and mixtures not realistic (and makes the discussion weak, e.g. lines 350-356).*

The comment is absolutely right and we are aware that species mixture composition depend on the fertilisation rate. Particularly legumes are supressed as it is the case in Mix4. We are totally aware that most benefits that derive from a biodiversity approach by allelopatic effects or plant microbiome interactions are highest under low or ideally no fertilization (e.g. Chaparro et al., 2012). We agree that fertilization of pure legumes is not necessary and also not recommended by us (see statement in discussion L316). However, we have to deal with the practical needs in farming reality. The basic decision to fertilize all treatments equally was the experience of weak established CC after soil depletion from the previous crop. Fertilization guarantees the better biomass development above and belowground and was in line with the farming practices. A major research goal of the project is the investigation of the CC potential for sustainable intensification of cropping systems. Thus, we needed an equal fertilisation strategy between all treatments to guarantee the comparability between the systems. Project recommendations should prove how much fertilizer N we can finally save by different CC treatments.

As the field trials was designed in 2015 the legal fertilization allowed the application rate of up to 60 kg total N fertilizer or 120 kg N as farmyard manure or compost. The common practice in lifestock farming systems is to empty the farm yard stocks before winter, when manure application is not allowed anymore. Cover crops are currently the only opportunity in legal regulations to apply fertilizers in the autumn. Even in "red areas" which are regions in Germany with high groundwater contamination by leached fertilizer N, up to 120 kg N can be applied as farmyard manure or compost to cover crops (independent if non-legume or legume). As consequence, farmers grow cover crops to get rid of the manure in autumn. For legumes, farmers have to compensate the N from legume fixation by reducing the N fertilization of the following crop by 10%. This reduction we have already included in the whole fertilization strategy across **all** CC treatments. The N fertilization of Maize was reduced by 10 % of the official recommendations. One of the major aims of this manuscript was to demonstrate that CC mixtures with legumes are equally efficient to deplete mineral soil N pools in autumn as non-legume CC and retain N over the winter. Further it should be demonstrated that pure legume CC (which are still allowed to fertilize) cannot compensate high N loads from autumn fertilization. We included a more careful discussion of the results and recommendations on fertilizer reduction e.g. L358.

We agree that different species have frost hardiness with different limits. In a mixture some components are standing longer than others. This is one of the advantages in mixtures. We address this issue in a small paragraph in the discussion part, highlighting the limitation of the data. Nevertheless, a short deep frost event (as is was the case last season in our field sites at 10[th] of December 2021) can kill the cover crops in one event and earlier than demanded. Latest to the end of January all CC in the experiment are frost killed reliably and laying on the ground (see Figures S5, S6).Than we still have to go 3 month with decaying biomass until next seeding. Also an early CC mulching by framers is quite frequent if CC are going to produce seeds. The termination of CC in our experiment is therefore important to quantify the potential of N mobilization from decomposing CC over the winter. We set a common starting point for the mineralization, which would be otherwise not monitored equally.

*The methodology needs to be clarified substantially. It is important to include information on soil sampling, CC sampling and termination in the main text. How many times did you take soil samples, 9 (Figure S2) or 10 (line 148)? Why do you take root samples down to 100 cm (then present only aggregated results to 90 cm), and soil samples to 80? In line 120-121, what do you mean by "ten randomly taken samples per soil depth"?*

We included some more information about sampling points in the main text. We sampled 9 times according to figure S2. We also add some more information on the plot design (L105-110).There was a mistake in the main text. Root samples were also summarized to 100 cm not 90 cm. The different sampling depth between roots and soil samples derived from different sampling approaches. Samples for root washing were taken by an automatic soil corer with 6 cm diameter equipped to a tractor. For root washing we needed much more soil volume than for $N_{min}$ extraction. Soil $N_{min}$ stocks was summarized to 80 cm soil depth. The four 10 cm increments we have chosen, because we wanted to monitor the $N_{min}$ development in certain depth. Otherwise we would get a mixed signal if we would have sampled 20 or 30 cm increments. With the interpolation approach we are able to model the missing increments and getting more precise information in a certain layer.

*In the modeling, why did you use a local polynomial regression model? How do you justify the modeled increase in soil Nmin between April 11 (soil sampling) and April 24 (first maize N fertilization) (see Figure 5)?*

A polynomial regression fits non-linear relationships. Loess models are very flexible making it possible to fit unknown relationships very well. Further, it allows us to fit a model without specification of unknown estimation parameters in a function. This enables us to predict the uncertainties in $N_{min}$ development between the time points more realistic than with simple linear models. The increase in sampling at April 11 and first maize fertilization is an artefact of the model. Actually it should be a more abrupt increase after fertilization. That was the reason why I plotted the sampling dates as asterisk in the graph. So the viewer can directly extract what is model and what is measured. We also tested different models (linear, spline, etc.) but none of them fitted the data appropriate. We also tried to replicate the data from sampling April 11 one day before fertilization. This would prevent the gradual increase but produced even more bias in a negative direction. So we assume the models at the present stage represents the most realistic modulation of the available data base. I additionally marked the fertilization events and add some information in the figure description to limit model assumptions.

*Your assessment of N leaching is based on monitoring of soil mineral N. It would be good to address other possible losses, and to discuss why you can assume that N losses from your system are not, for example, as N2O.*

The comment is totally right. Gaseous losses are also important for cover cropping periods particularly for legumes (Basche et al., 2014; Abdalla et al., 2019; Bodner et al., 2017). We also have a full data set on greenhouse gas monitoring over the observation period (CO2, CH4, and NH3). However due to the focus on Nmin and soil water we do not wanted to extend the manuscript further. Discussion of greenhouse gasses was planned as part of a second manuscript together with microbial biomass data. I included, however, a small paragraph on this issue in the discussion part (L 340-347). Based on our data we found N2O emissions coupled to precipitation periods (see graph below). The general view is that different CC treatments vary in their temporal N2O emission dynamic but were not substantially different from the fallow treatments. But I have not evaluated the cumulative fluxes and statistic differences so far. If the editor and the reviewer deciding, that the data are indispensable for the story, I will include them. But this will require further time to include new sections in material and methods, results, discussion and do statistical analyses, prepare data and R scripts for upload. This will extend the volume of the manuscript, which I tried to keep comprehensive so far.

[Figure]

*2019 was not a dry year (Figure S1 and line 94), but this is not taken into account in the discussion. In addition, line 25 cannot be a conclusion from your study.*

We agree that not the whole year 2019 was a dry year on our field site and corrected the statement in the abstract. Despite this the water deficit in summer 2019 was high and water availability for maize growth reduced as compared to normal years. In the table below it is clearly visible that the winter 2019 was exceptionally dry, the spring brought higher precipitation than average but the summer 2019 was dry again. Also our data logger and soil measurements suggested that soil moisture was below wilting point (~13 %vol) already to the end of July. This is also supported by ~ 10 dt lower yield levels as compared to normal years. Thus, water availability was a limiting factor in 2019. See also the new Table S5.

| Year | Meteorological year (mm) | Winter (mm) | Spring (mm) | Summer (mm) | Autumn (mm) |
|---|---|---|---|---|---|
| 2015 | 882.8 | 241.9 | 132.6 | 276 | 294 |
| 2016 | 813.4 | 280.8 | 147.4 | 268.2 | 125.4 |
| 2017 | 1192.2 | 250 | 141.8 | 354.6 | 370.6 |
| 2018 | 535 | 274.4 | 111.4 | 80.2 | 110.2 |
| 2019 | 779.8 | 188.4 | 234.2 | 144 | 242.2 |
| 2020 | 684.8 | 256.4 | 91.4 | 203.8 | 124.8 |

*Your conclusions on the effect of CC on growing conditions for the following crop are not based on maize growth (and this is not even discussed). If you have data, it would be a very good idea to include them.*

We agree that we cannot draw conclusions on maize growth without showing evidence. Yield data was planned to be integrated in another publication of our colleagues from agronomy.

My colleagues agreed that we can show the data from 2019 harvest (now Fig. 6). I also included a paragraph in the method section, results and discussion. As our colleagues were responsible for data collection and management of agronomic parameters we have to include two more co-authors to the list: Robin Kümmerer and Bernhard Bauer from University of Applied Science Weihenstephan-Triesdorf.

*Technical comments:*

*Please do a careful revision of the language (especially in the discussion).*

We checked language again carefully and corrected a few typos. The manuscript was send for professional correction in advance.

*Line 15: change to "and two catch crop mixtures with 4 and 12 plant species (Mix4 and Mix12) with a fallow treatment"*

Done

*Lines 31-40: this can be shortened*

Done

*Lines 45-49: what about legumes and mixtures with legumes?*

Can be fertilized as well with some restrictions in "red areas" (see response to first comment). I add a sentence for clarification.

*Line 56: redundancy of what?*

Changed to multifunctionality

*Lines 58-59: what do you mean by this?*

Microbial utilization is a considerable sink for mineral N pools and lead to their immobilization. Stimulating microbial biomass is thus and additional pathway to reduce leaching. Our data on Microbial biomass (will be part of another manuscript) indicated that up to 40 kg N ha$^{-1}$ can be incorporated by microbial biomass over the winter.

*Line 64: change from "measure for" to "indicator of"*

Changed.

*Lines 64-67: please rephrase (this is not the discussion)*

We do not agree with this. The C:N ratio is part of the manuscript, since we measured, showed and discussed at least partially the impact of C:N ratio on winter mineralization. We

assume that pointing to the importance of litter quality in the introduction is important to get into the topic.

*Lines 68-71: there are a few more studies on "the potential for the reduction of winter N leaching losses of CC mixtures in comparison to single species", and I encourage the authors to make one more search.*

Yes that is right. We add some more references and corrected the statement. There are quite some new works since I started to write the manuscript.

*Lines 78-79: I think the objectives of the study should be phrased in relation to the results. E.g., the objective is to determine the effect of something.*

We clarified the objectives.

*Line 103: Vicia pannonica is a legume.*

Changed.

*Lines 224-234: I am not sure I understand this. Can you explain and discuss it (also in connection to other possible N losses)?*

For the example of fallow, we detected three main leaching events visible in Fig. 5. Nmin to 80cm soil depth increased to ~121 kg ha$^{-1}$ dropped to 64 (event 1) increased again to 88kg dropped to 71kg (event 2), increased to 76kg dropped to 48kg (event 3). The sum of these events is 102 kg. The exact calculation steps are provided in the R file and can be recalculated from the data. For mustard this was not possible, because Nmin stocks were constantly rising. The constant mineralization and diffuse migration towards the subsoil resulted in constant increase of Nmin in the subsoil. We assume a linear transport towards the 70-80 cm layer and the same constant transport further down the profile. We calculated the daily lateral transport rate based on the linear regression function. Based on this linear function we calculated a the daily transport rate towards the 70-80 cm layer. We assume that lateral transport will not stop at this layer and we will have the same flow rate below the 80 cm soil depth.

The leaching events as described above correspond well with precipitation pattern and thus with N2O emissions (plot above). But without showing N2O data it is too speculative to go deeper to into discussion about other possible losses.

*Lines 281-281: what does this mean?*

33% about fallow level. I added this information.

*Lines 283-287: what about maize growth? Could it explain some of this?*

No. We added silage maize yield levels. Clover was not different to oat or phacelia.

*Lines 309-310: why?*

Energy is saved for synthetic N fertilizer production, transport, reduced passages with machinery.

*Lines 311-313: unclear sentence*

Rewritten.

*Line 316: what previous studies?*

One of our previous greenhouse experiments, see Table S4.

*Line 334: "…have been found to determine a higher carbon-use efficiency"?*

Changed.

*Lines 336-337: unclear sentence (and see references suggested above)*

The sentence was corrected and statements revised.

*Conclusion: too long. All the first paragraph can be removed. The specific comments above should be taken into account also here.*

We erased the first paragraph and added some conclusions based on the revised comments.

- *Figures: check that the text is in English, add a line for CC termination (figures 2, 3, 4 and 5). It would be good if dates in the x axis were the sampling dates (where relevant).*

We added the information as suggested. Sampling dated were marked by asterisk.

- *Table 2: what do capital letters represent?*

I do not know why capital letters appeared. It should not be the case and was not part of my original table. I corrected it again.

**References**

Abdalla, M., Hastings, A., Cheng, K., Yue, Q., Chadwick, D., Espenberg, M., Truu, J., Rees, R. M., and Smith, P.: A critical review of the impacts of cover crops on nitrogen leaching, net greenhouse gas balance and crop productivity, Glob. Change Biol., 25, 2530–2543, https://doi.org/10.1111/gcb.14644, 2019.

Basche, A. D., Miguez, F. E., Kaspar, T. C., and Castellano, M. J.: Do cover crops increase or decrease nitrous oxide emissions? A meta-analysis, J. Soil Water Conserv., 69, 471–482, https://doi.org/10.2489/jswc.69.6.471, 2014.

Bodner, G., Mentler, A., Klik, A., Kaul, H.-P., and Zechmeister-Boltenstern, S.: Do cover crops enhance soil greenhouse gas losses during high emission moments under temperate Central Europe conditions?, Bodenkult. J. Land Manag. Food Environ., 68, 171–187, https://doi.org/10.1515/boku-2017-0015, 2017.

Chaparro, J. M., Sheflin, A. M., Manter, D. K., and Vivanco, J. M.: Manipulating the soil microbiome to increase soil health and plant fertility, Biol. Fertil. Soils, 48, 489–499, https://doi.org/10.1007/s00374-012-0691-4, 2012.